# The Effects of TiO_2_ Nanoparticles on Cisplatin Cytotoxicity in Cancer Cell Lines

**DOI:** 10.3390/ijms21020605

**Published:** 2020-01-17

**Authors:** Basma Salama, El-Said El-Sherbini, Gehad El-Sayed, Mohamed El-Adl, Koki Kanehira, Akiyoshi Taniguchi

**Affiliations:** 1Cellular Functional Nano Biomaterials Group, Research Center for Biomaterials, National Institute for Materials Science (NIMS), 1-1 Namiki, Tsukuba, Ibaraki 305-0044, Japan; dr.basma_bio@yahoo.com (B.S.); koki.kanehira@jp.toto.com (K.K.); 2Department of Biochemistry and Chemistry of Nutrition, Faculty of Veterinary Medicine, Mansoura University, 60 El Gomhouria St., Mansoura, Dakahlia Governorate 35516, Egypt; sshrbini@mans.edu.eg (E.-S.E.-S.); grelsayed@yahoo.com (G.E.-S.); drmohamedalymaher@hotmail.com (M.E.-A.); 3Inorganic Materials Research Section, TOTO Ltd. Research Institute, Honson 2-8-1, Chigasaki, Kanagawa 253-8577, Japan

**Keywords:** titanium dioxide nanoparticles, cisplatin, cytotoxicity, drug resistance, P-glycoprotein

## Abstract

There have been many studies on improving the efficacy of cisplatin and on identifying safe compounds that can overcome multi-drug resistance (MDR) acquired by cancer cells. Our previous research showed that polyethylene glycol-modified titanium dioxide nanoparticles (TiO_2_ PEG NPs) affect cell membrane receptors, resulting in their aggregation, altered localization and downregulation. TiO_2_ PEG NPs may affect P-glycoprotein (P-gp), a membrane efflux channel involved in MDR. In this study, we investigated the effect of TiO_2_ PEG NPs on cisplatin cytotoxicity. We used HepG2 cells, which highly express P-gp and A431 cells, which show low expression of P-gp. The results showed that 10 µg/mL 100 nm TiO_2_ PEG NPs increased intracellular cisplatin levels and cytotoxicity in HepG2 cells but not in A431 cells. TiO_2_ PEG NPs treatment decreased the expression level of P-gp in HepG2 cells. Our findings indicate that TiO_2_ PEG NPs enhance cisplatin cytotoxicity by down regulating P-gp and that TiO_2_ PEG NPs are promising candidates for inhibiting P-gp and reversing drug resistance acquired by cancer cells.

## 1. Introduction

Cisplatin is one of the most effective and widely used chemotherapeutic agents in the treatment of solid tumors, including bladder, lung, head and neck, ovarian and testicular cancers [1]. However, intrinsic or acquired drug resistance is the major obstacle in the use of cisplatin for chemotherapy [2]. Multi-drug resistance (MDR) is an important phenomenon in which many types of cancer cells acquire resistance against a broad range of structurally distinct anticancer drugs [3]. P-glycoprotein (P-gp), also called multi-drug resistance protein 1 (MDR1), is considered the most important membrane efflux pump involved in MDR [4]. The structure of P-gp, and and its traffic and cycling pathways between cellular compartments, are key to understanding the mechanism of MDR. P-gp is a trans-membrane protein with a glycosylated extracellular loop, 12 trans-membrane domains (TMDs) and two intracellular nucleotide-binding domains (NBDs) [5]. The TMDs form channels for the efflux of substrate drugs, whereas the NBDs are exposed to the cytoplasm and participate in ATP binding and hydrolysis [6]. In the presence of a substrate drug, P-gp undergoes conformational changes and the TMDs reorganize to act as channels for translocation of the drug out of the cell; in addition, the distance between the two NBDs decreases, leading to ATP hydrolysis [7]. Many studies have investigated the localization and trafficking of P-gp and have shown that P-gp is mainly localized on the cell membrane as a drug efflux pump. However, P-gp is also localized in many intracellular compartments, such as endoplasmic reticulum, Golgi, endosomes and lysosomes and/or proteasome [8,9]. In cell lines that express P-gp, trafficking from the endoplasmic reticulum and the Golgi apparatus is transient and P-gp is rapidly transported to the cell membrane by endosomes, explaining why P-gp is primarily localized on cell surfaces rather than in the cytoplasm [10]. It was previously reported that P-gp has a relatively long half-life on the cell membrane (14–17 h), followed by lysosomal degradation, and several compounds could affect the half-life of P-gp and increase its internalization by lysosomes [11].

There has been much effort recently to test and develop numerous natural and synthetic compounds that inhibit P-gp-mediated drug efflux and thus re-sensitize cancer cells to chemotherapeutics [12]. These sensitizers include calcium channel blockers, immunosuppressants, calmodulin antagonists and piperine [13,14]. Unfortunately, clinical trials of most of these compounds have shown them to be toxic, non-specific to P-gp and interacting with other cellular transporters, or interacting with cellular enzymes and affecting the metabolism of chemotherapeutic drugs [15,16]. P-gp inhibitors should be of low molecular weight, lipophilic, non-toxic and have no effect on cell metabolism. Moreover, potent P-gp inhibitors should interact with the P-gp TMDs or NBDs, leading to conformational change, loss of P-gp function and inducing P-gp lysosomal degradation [17,18].

Polyethylene glycol-modified TiO_2_ nanoparticles (TiO_2_ PEG NPs) are widely used in cancer imaging and drug delivery [19,20]. Much recent research on the interaction and effect of TiO_2_ PEG NPs on cancer cells has focused on improving their utility in nanomedicine [21]. We recently showed that TiO_2_ PEG NPs affect cell membrane receptors, resulting in their conformational change and altered localization. This interaction between TiO_2_ PEG NPs and the cell membrane modifies cell behavior and response to other chemical stimuli [22], suggesting that TiO_2_ PEG NPs may affect the membrane channel function of P-gp and thus modulate cisplatin cytotoxicity.

The aim of our study was to investigate the effect of TiO_2_ PEG NPs on cisplatin cytotoxicity. We used two cancer cell lines: The HepG2 cell line, which expresses a high level of P-gp, and the A431 cell line, which expresses a low level of P-gp. Our results indicated that TiO_2_ PEG NPs changed the localization and expression of P-gp and increased intracellular cisplatin accumulation and cytotoxicity in HepG2 cells, but had no effect on cisplatin cytotoxicity in A431 cells. These findings suggest that TiO_2_ PEG NPs hold promise as a P-gp inhibitor to reverse MDR and increase chemotherapeutics efficacy. 

## 2. Results

### 2.1. IC50 of Cisplatin in HepG2 and A431 Cells

We evaluated the IC50 of cisplatin in HepG2 and A431 cells by treating the cells with serial dilutions of cisplatin for 24 h, then staining with calcein AM for live cells and with EthD-1 for dead cells. The IC50 values were calculated from a dose–response curve. The estimated IC50 values were 12 µg/mL and 6 µg/mL for HepG2 and A431 cells, respectively.

### 2.2. The Effect of TiO_2_ PEG NPs on Cisplatin Cytotoxicity in HepG2 and A431 Cell Lines

We recently showed that TiO_2_ PEG NPs affect cell membrane receptors, resulting in receptor aggregation, changes in localization and modulation of cellular behavior [22]. The results suggested that TiO_2_ PEG NPs may affect the function of P-gp and thus here, we investigated the effect of TiO_2_ PEG NPs on cisplatin cytotoxicity. Cellular cytotoxicity was assayed to evaluate the effect of different concentrations of 100 nm, 200 nm and 300 nm TiO_2_ PEG NPs on cisplatin cytotoxicity in HepG2 and A431 cells.

As shown in Figure 1A, all three sizes of TiO_2_ PEG NPs increased HepG2 cell viability compared to control untreated cells. The lowest concentration (10 µg/mL) of the smallest NPs (100 nm) had the most significant effect and increased cell viability up to 160% compared to the control, although the largest NPs (300 nm) still significantly increased cell viability, up to 120%. We previously showed that 100 nm TiO_2_ PEG NPs induced cell proliferation at low concentration and that TiO_2_ PEG NPs induce hepatocyte growth factor receptors (HGFRs) aggregation, thereby inducing cell proliferation [22]. Similar trends were observed with A431 cells (Figure 1C). Our results indicate that different sizes of TiO_2_ PEG NPs have no cytotoxic effect on either cell lines and that smaller NPs (100 nm) at low concentration in particular induce HepG2 and A431 cells proliferation.

We evaluated the effect of TiO_2_ PEG NPs on cisplatin cytotoxicity by treating HepG2 and A431 cells with the IC50 of cisplatin mixed with different concentrations of 100 nm, 200 nm and 300 nm TiO_2_ PEG NPs. As shown in Figure 1B, without TiO_2_ PEG NPs, the viability of cisplatin-treated HepG2 cells decreased to 50% compared to control untreated cells. A mixture of TiO_2_ PEG NPs with the IC50 of cisplatin significantly decreased HepG2 cells viability compared to cisplatin-treated cells, without TiO_2_ PEG NPs, and the lowest concentration (10 µg/mL) of small NPs (100 nm) was most effective, decreasing cell viability from 50% to 20%. A431 cells treated with mixture of TiO_2_ PEG NPs and the IC50 of cisplatin showed increased viability (Figure 1D), these results were similar to the results obtained when cells were treated only with TiO_2_ PEG NPs (Figure 1C), suggesting that TiO_2_ PEG NPs had no effect on cisplatin cytotoxicity in A431 cells. These results indicated that 100 nm TiO_2_ PEG NPs, increased cisplatin cytotoxicity in HepG2 cells, whereas TiO_2_ PEG NPs had no effect on cisplatin cytotoxicity in A431 cells.

### 2.3. TiO_2_ PEG NPs Enhance Cisplatin Uptake by HepG2 Cells

We investigated the molecular mechanism underlying the effect of TiO_2_ PEG NPs on cisplatin cytotoxicity by assessing intracellular cisplatin levels in HepG2 and A431 cells from their intracellular platinum concentrations. Cells were treated with the IC50 of cisplatin with or without 100 nm TiO_2_ PEG NPs for 24 h, then 10^6^ cells were digested and the intracellular platinum concentrations were measured by ICP-MS.

As shown in Figure 2A, HepG2 cells treated with mixture of cisplatin and TiO_2_ PEG NPs showed increased intracellular cisplatin level compared with cisplatin-treated cells. In contrast, TiO_2_ PEG NPs had no effect on the intracellular cisplatin levels of A431 cells (Figure 2B). These results indicate that TiO_2_ PEG NPs increase the intracellular cisplatin concentration in HepG2 cells but not in A431 cells.

### 2.4. Cellular Uptake of TiO_2_ PEG NPs

We investigated whether TiO_2_ PEG NPs affect the cell surface or intracellular metabolism by measuring the cellular uptake of different concentrations of 100, 200 and 300 nm TiO_2_ PEG NPs by HepG2 and A431 cells using flow cytometry. As shown in Figure 3A, TiO_2_ PEG NPs uptake increased as the size and concentration of the NPs increased. Similar trends were observed with A431 cells (Figure 3B). These results indicated that the uptake of TiO_2_ PEG NPs by both cell lines is size- and dose-dependent. In case of low concentration (10 µg/mL) and small TiO_2_ PEG NPs (100 nm), percentage of cells taking up NPs was very low, indicating that most of TiO_2_ PEG NPs were outside the cells. The free NPs in the culture medium could not affect the cellular response. This suggested that cell surface NPs could possibly affect cells surface proteins that play key roles in cisplatin cytotoxicity. These surface proteins could be primarily expressed in HepG2 cells not by A431 cells.

### 2.5. Localization of P-gp by Immunofluorescence Staining

P-gp is a surface protein responsible for the efflux of cisplatin and other chemotherapeutic agents to outside the cell, leading to decreased intracellular drug accumulation and thus inducing MDR. We investigated how TiO_2_ PEG NPs affect P-gp by investigating its localization and expression. HepG2 and A431 cells were treated with cisplatin with or without 100 nm TiO_2_ PEG NPs, then the localization of P-gp was visualized by immunofluorescence staining. Confocal microscopy images showed that P-gp was highly expressed on HepG2 cell surface. P-gp in HepG2 cells treated with TiO_2_ PEG NPs and cisplatin was mainly localized inside the cells rather than on the cells surface (Figure 4A). The same results were obtained in different focus pictures. In contrast, little P-gp was visualized in A431 cells and TiO_2_ PEG NPs had no effect on P-gp expression and localization (Figure 4B). Figure 4C,D shows quantitative calculated results of P-gp expression levels. The expression of P-gp was significantly decreased by TiO_2_ PEG NPs treatment in HepG2 cells, but not in A431 cells. These results indicated that changes in P-gp localization and expression in HepG2 cells increased cisplatin cytotoxicity. TiO_2_ PEG NPs altered the localization of P-gp to the cytoplasm of HepG2 cells, resulting in decreased expression in the cell membrane, loss of P-gp function and cisplatin retention inside the cells.

## 3. Discussion

Drug resistance of cancer cells against a wide range of drugs, including cisplatin, is a major obstacle in cancer chemotherapy and there has been much effort to develop compounds that can sensitize cancer cells towards chemotherapeutic drugs. Unfortunately, most of these chemosensitizers have proven inadequate and thus, in this investigation we studied the effect of TiO_2_ PEG NPs on cisplatin cytotoxicity.

We found that low concentrations of 100 nm TiO_2_ PEG NPs increased HepG2 and A431 cells viability. Our previous studies concluded that nanoparticles can interact with cell membrane receptors, leading to receptors aggregation, change in receptors localization and in modulation of receptors expression. We also previously found that low concentrations of TiO_2_ PEG NPs induced aggregation of hepatocyte growth factor receptors (HGFRs) in HepG2 cells and induced cell proliferation [22]. Moreover, polystyrene NPs induced aggregation of epidermal growth factor receptors (EGFRs) in A431 cells [23]. In addition, we showed that 200 nm silver NPs reduced lung epithelial cell surface expression of tumor necrosis factor receptor 1 (TNFR1) with increased localization of receptors in the cell cytoplasm [24]. These results suggested that NPs affect cell surface protein localization and expression. In this paper, we observed that TiO_2_ PEG NPs affected P-gp localization and expression.

Previous papers confirmed that interactions between P-gp and inhibitors lead to P-gp conformational changes that interfere with TMDs channel formation, changes in the distance between NBDs and inhibit NBDs ATPase activity, subsequently leading to lysosomal degradation [15,25]. Similarly, we suggested that TiO_2_ PEG NPs can interact with the function of P-gp as a membrane channel and inhibit its drug efflux activity.

A possible molecular mechanism for the effect of TiO_2_ PEG NPs on cisplatin cytotoxicity is illustrated in Figure 5. We propose that TiO_2_ PEG NPs associate with the TMDs of P-gp and interfere with their re-organization to form channels for drug efflux. Moreover, TiO_2_ PEG NPs induce conformational changes that could affect the distance between the NBDs, leading to inhibition of their ATPase activity. Finally, the interaction between TiO_2_ PEG NPs and P-gp induces P-gp degradation and increases intracellular cisplatin levels and cytotoxicity.

## 4. Materials and Methods

### 4.1. Cell Lines and Cell Culture

We used the HepG2 cell line, derived from hepatic cell carcinoma, and the A431 cell line, derived from epithelial cell carcinoma. HepG2 and A431 cells were cultured at 37 °C and 5% CO_2_ in High Glucose Dulbecco’s modified Eagle medium (DMEM, high glucose, Nacalai Tesque, Kyoto, Japan) supplemented with 10% (*v/v*) heated fetal bovine serum (Biowest, Riverside, MO, USA), 100 μg/mL penicillin and 10 μg/mL streptomycin (Nacalai Tesque). Cells were sub-cultured every two days at 70–80% confluency.

### 4.2. Preparation of Nanoparticles 

TiO_2_ NPs were supplied by Fuji Chemical Co., Ltd. (Osaka, Japan) and surface modification with PEG was conducted as described previously [26]. Briefly, 100 mL solution of 0.1 M titanium ethoxide in ethanol with 50 % (*v/v*) acetonitrile was mixed and hydrolyzed for 60 min at room temperature, with the addition of ammonium hydroxide in the final concentration range of 0.01–0.1% (*w/v*), depending on the desired particle size (100, 200 or 300 nm). After hydrolysis, the mixed solution was heated at 80 °C for 3 h under reflux. The surface of the spherical TiO_2_ particles was then coated with a PEG co-polymer. Finally, TiO_2_ PEG NPs (100 nm, 200 nm and 300 nm diameter) were centrifuged at 14,000× *g* for 30 min and suspended in endotoxin-free sterilized water three times. The hydrodynamic particle sizes of the TiO_2_ PEG NPs in water were measured by dynamic light scattering (zeta sizer nano ZS, Malvern, UK) and were 126.6 nm, 210.9 nm and 282.7 nm for the 100 nm, 200 nm and 300 nm TiO_2_ PEG NPs, respectively.

### 4.3. Preparation of Cisplatin

Cisplatin (cis-diamminedichloroplatinum (II)) was obtained from Fujifilm (Wako Ltd., Osaka, Japan). A 0.5 mg/mL stock solution prepared by dissolving cisplatin powder in 0.9 M NaCl solution was stored at 4 °C and used within one week. The desired final concentrations were obtained by diluting the stock solution with culture medium.

### 4.4. Cell Viability/Cytotoxicity Assay

First, the half maximal inhibitory concentration value (IC50) of cisplatin for HepG2 and A431 cells were evaluated using LIVE/DEAD^®^ Viability/Cytotoxicity Kit (Invitrogen, Ltd., Cambridge, UK) according to the manufacturer instructions by fluorescence microplate Protocol. Briefly 1 × 10^4^ cells/well were seeded in costar 96 well plate and incubated at 37 °C and 5% CO_2_. After 24 h, cells were exposed to different concentrations of cisplatin for another 24 h. Finally, cells were stained by 1 µM calcein AM for live cells and 2 µM ethidium homodimer-1 (EthD-1) for dead cells and fluorescence intensity was measured by Spark™ 10 M multimode microplate reader (Tecan Ltd., Männedorf, Switzerland). IC50 was determined from the dose–response curve. The mean IC50 value of cisplatin for each type of cell lines were calculated from three independent experiments each of which were performed in triplicate.

After that, the effect of TiO_2_ PEG NPs on HepG2 cells and A431 cells viability and cisplatin cytotoxicity was assessed according to the same protocol using LIVE/DEAD^®^ Viability/Cytotoxicity Kit (Invitrogen). Briefly, 1 × 10^4^ cells/well were seeded in costar 96 well plate and incubated at 37 °C and 5% CO_2_. After 24 h cells were exposed to 100 nm, 200 nm or 300 nm TiO_2_ PEG NPs in different concentrations (0, 10, 40, 100, 400 µg/mL medium) with or without IC 50 of cisplatin for 24 h and the fluorescence intensity was measured by Spark™ 10 M multimode microplate reader (Tecan) after staining the cells by1µM calcein AM for live cells and 2 µM EthD-1 for dead cells.

### 4.5. Evaluation of Cellular Cisplatin Uptake by ICP-MS Analysis

Cisplatin uptake by HepG2 and A431 cells was assessed by measuring intracellular platinum concentration using inductively coupled plasma mass spectrometry (ICP-MS). Briefly, 10^6^ cells/well were seeded in 24 well plates and incubated at 37 °C and 5% CO_2_. After 24 h, the cells were exposed to the IC50 of cisplatin either alone or mixed with 10 µg/mL 100 nm TiO_2_ PEG NPs. After 24 h, the cells were washed with PBS, collected by trypsinization, washed twice with PBS and counted following trypan blue staining. Cells (10^6^) cells were suspended in 1 mL PBS and stored at 4 °C until digestion. Samples were transferred to a quartz beaker, then 1.5 mL HCl and 0.5 mL HNO_3_ were added and the samples were dried by heating. After cooling, a further 1.5 mL HCl and 0.5 mL HNO_3_ were added to complete digestion by heating, then the obtained solutions were poured into a polypropylene vessel and diluted to 25 mL with Milli-Q water. A calibration curve was constructed using standard solutions prepared by serial dilution of a reference solution (Kanto Chemical Co, Inc., Tokyo, Japan) and sample platinum concentrations were measured using the Pb internal standard method by ICP-MS. Cellular cisplatin uptake was expressed as (µg/10^6^ cells).

### 4.6. Evaluation of Cellular Uptake of TiO_2_ PEG NPs by Flow Cytometry

The percent of HepG2 and A431 cells incorporated with TiO_2_ PEG NPs was assessed depending on changes in light scattering using flow cytometry. Forward scatter (FSC) is the laser light scattered at narrow angles to the axis of the laser beam and is proportional to the cell size. Side scatter (SSC) is the laser light scattered at a 90° angle to the axis of the laser and is proportional to the intracellular granularity, which is increased by the uptake of nanoparticles. Briefly, 10^6^ cells/well were seeded in 24 well plates and incubated at 37 °C and 5% CO_2_ for 24 h, then the cells were exposed to different concentrations (0, 10, 40, 100, 400 µg/mL medium) of 100 nm, 200 nm and 300 nm TiO_2_ PEG NPs. After 24 h, the cells were washed twice, collected by trypsinization, washed three times with PBS, dispersed in 1 mL of 6% heated fetal bovine serum in phosphate buffer saline (HFBS/PBS) solution and stored on ice and analyzed within one hour. Immediately prior to analysis, the cells were passed through a nylon mesh (Cell Strainer Snap Cap, Falco, NY, USA), then cellular internal granularity was assessed using SSC and cell size was assessed using FSC using a SP6800 spectral analyzer (Sony Biotechnology, Tokyo, Japan). The percentage of cells containing nanoparticles was calculated based on changes in the gated areas compared with control untreated cells.

### 4.7. Immunofluorescence Staining of P-gp

HepG2 and A431 cells were seeded in 4-compartment cell view cell culture dishes (Greiner Bio-One, Inc., Monroe, NC, USA) at a density of 4 × 10^4^ cells/compartment at 37 °C and 5% CO_2_ for 24 h. Next, the cells were treated with the IC50 of cisplatin with and without 10 µg/mL 100 nm TiO_2_ PEG NPs for 24 h. The cells were washed with PBS and fixed with 4% paraformaldehyde (PFA) for 10 min, then permeabilized with 0.1% Triton X-100. Subsequently, the cells were blocked with 1% bovine serum albumin (BSA)/10% normal goat serum/0.3 M glycine in 0.1% Tween-PBS for 1 h at room temperature, then incubated with anti P-gp antibody (1/500 dilution, Abcam, London, UK) at 4 °C overnight. The cells were washed three times (10 min each) with PBS and incubated with goat anti-mouse IgG H&L (1/500 dilution, Abcam) in the dark for 1 h at room temperature, followed by washing three times (10 min each) with PBS. Nuclear DNA was labeled with DAPI (Thermo Fisher Scientific, Waltham, MA, USA). Images were taken using a confocal laser scanning microscope (LSM510 META, Carl Zeiss Inc., Jena, Germany). Fluorescence intensity of Alexa Fluor 488 was measured by ImageJ program [27].

### 4.8. Statistical Analysis

All data were assessed for statistical significance using Student’s *t*-test. All values are presented as mean ± SD with three or more independent replicates (*n* ≥ 3). ** *p* ≤ 0.01, which is indicated in the figure legends.

## 5. Conclusions

In this paper we showed that TiO_2_ PEG NPs increase cisplatin cytotoxicity towards P-gp-expressing cells by inhibiting P-gp. Our results suggest that TiO_2_ PEG NPs are promising candidates for reversing MDR and inhibiting P-gp functions in cancer cells. Our findings indicate that low concentrations of TiO_2_ PEG NPs could efficiently inhibit P-gp function; this point would be an important advantage for safety use that could promote TiO_2_ PEG NPs for clinical trials.

## Figures and Tables

**Figure 1 ijms-21-00605-f001:**
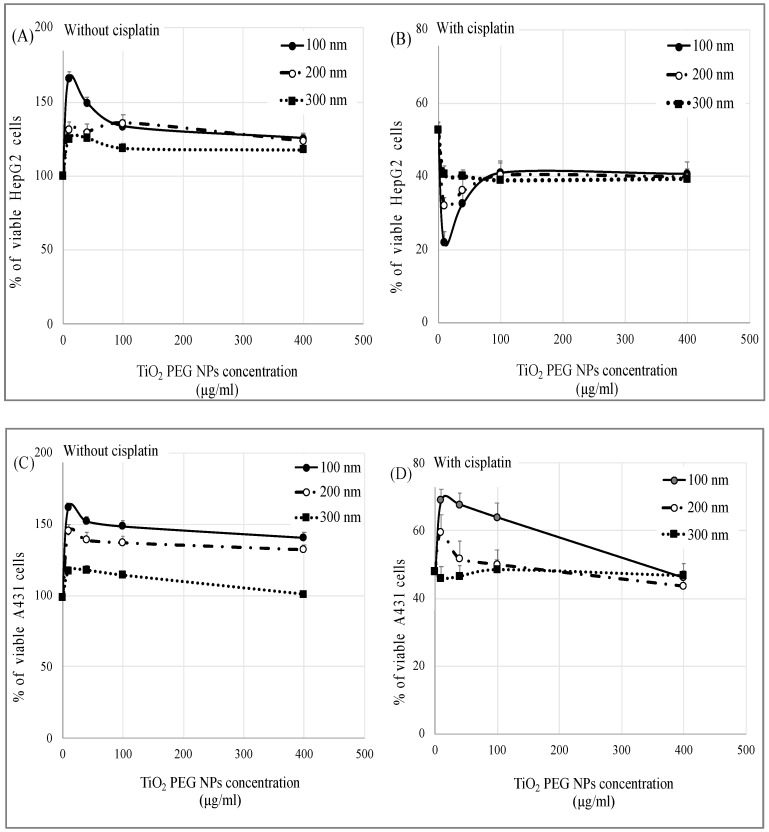
Effect of polyethylene glycol-modified titanium dioxide nanoparticles (TiO_2_ PEG NPs) on cisplatin cytotoxicity. HepG2 (**A**,**B**) and A431 cells (**C**,**D**) were exposed to different concentrations of 100 nm (closed circles), 200 nm (open circles) and 300 nm (closed rectangles) TiO_2_ PEG NPs for 24 h in the presence (**B**,**D**) or absence (**A**,**C**) of the IC50 of cisplatin. All values are normalized to control untreated cells. All values are presented as mean ± SD (*n* ≥ 3). Data were analyzed using Student’s *t*-test.

**Figure 2 ijms-21-00605-f002:**
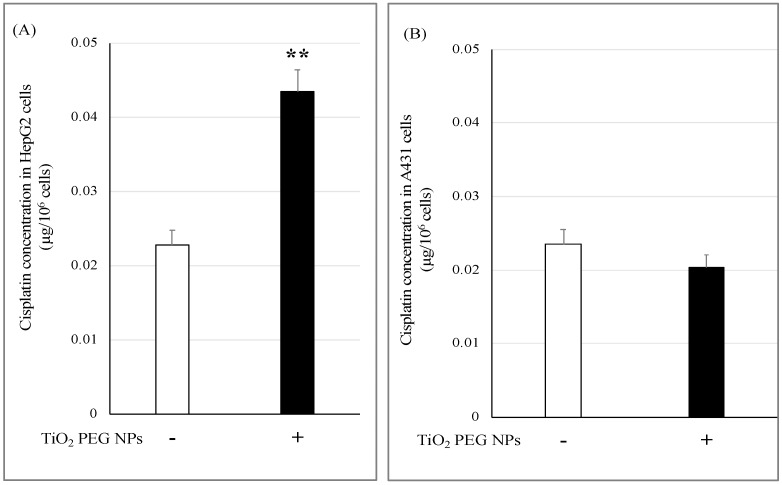
TiO_2_ PEG NPs enhance cisplatin uptake by HepG2 cells. HepG2 (**A**) and A431 cells (**B**) were treated with the IC50 of cisplatin with (black bars) or without (white bars) 10 µg/mL 100 nm TiO_2_ PEG NPs for 24 h. A calibration curve was constructed using standard platinum dilutions of a reference solution and the correlation coefficient (R^2^) was 1.0. All values are presented as mean ± SD (*n* ≥ 3). Data were analyzed using Student’s *t*-test; ** *p* ≤ 0.01.

**Figure 3 ijms-21-00605-f003:**
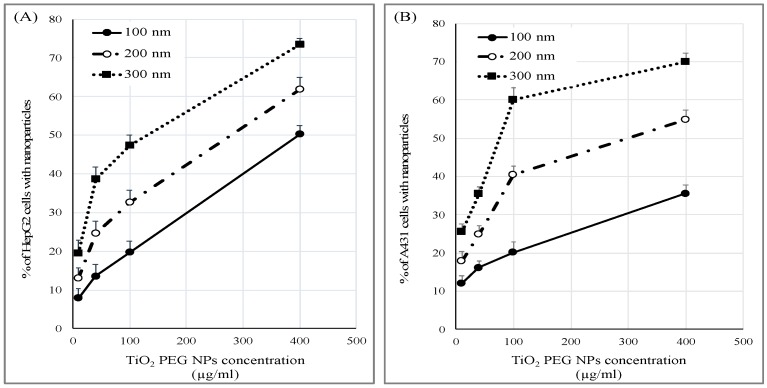
Size- and dose-dependent uptake of TiO_2_ PEG NPs by cancer cell lines. HepG2 (**A**) and A431 cells (**B**) were exposed to different concentrations of 100 nm (closed circles), 200 nm (open circles) or 300 nm TiO_2_ PEG NPs (closed rectangles) for 24 h. Cellular NPs uptake efficacy was normalized to control untreated cells. All values are presented as mean ± SD (*n* ≥ 3). Data were analyzed using Student’s *t*-test.

**Figure 4 ijms-21-00605-f004:**
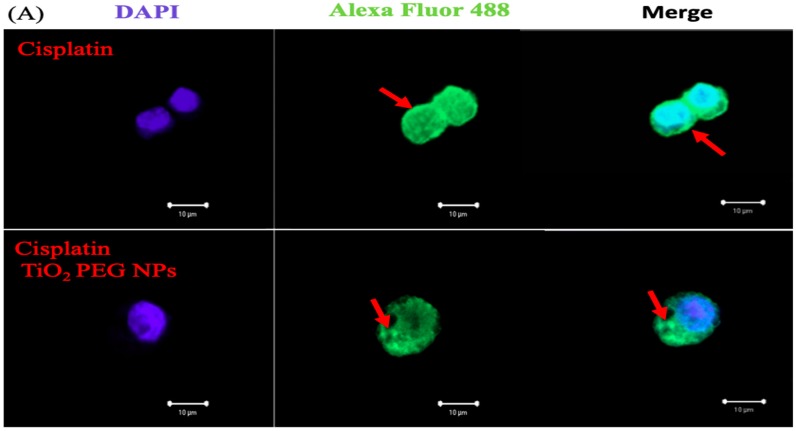
TiO_2_ PEG NPs change P-gp localization and expression in HepG2 cells. HepG2 (**A**) and A431 (**B**) cells were treated with the IC50 of cisplatin without (upper photos) or with (lower photos) 10 µg/mL 100 nm TiO_2_ PEG NPs for 24 h, followed by immunofluorescence staining with anti-P-gp antibody. The fluorescence intensities of Alexa Fluor 488 were calculated in HepG2 (**C**) and A431 cells (**D**) exposed to cisplatin with (black bars) or without (white bars) TiO_2_ PEG NPs. All values in (**C**,**D**) are presented as mean ± SD (*n* ≥ 3). Data were analyzed using Student’s *t*-test; ** *p* ≤ 0.01.

**Figure 5 ijms-21-00605-f005:**
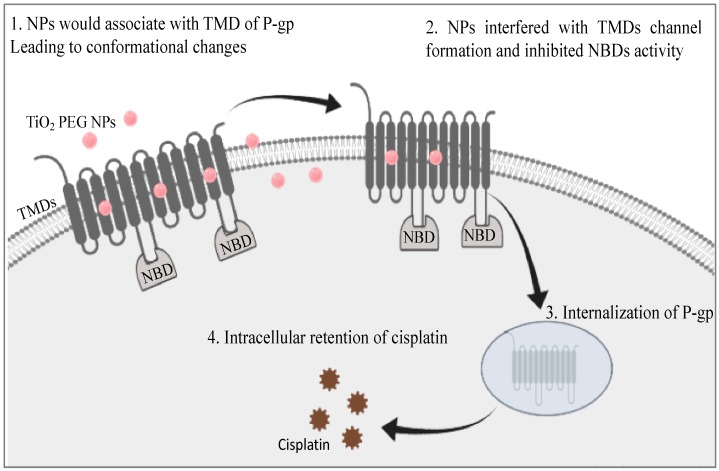
Proposed molecular mechanism for the effect of TiO_2_ PEG NPs on cisplatin cytotoxicity in HepG2 cells by the downregulation of P-gp.

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
