# Peer review of "The Effects of TiO2 Nanoparticles on Cisplatin Cytotoxicity in Cancer Cell Lines"

_ijms, 2020, doi:10.3390/ijms21020605_

Round 1

Reviewer 1 Report

The authors should describe the methods more completely. It is often hard to get the full information that would allow to reproduce the experiments. One example is item 4.2, where they state preparation of TiO2 PEG NP was described previously. They should describe the preparation briefly. Item 4.4 presents the same problem mentioned in comment 1. Line 222 says “assessed using a LIVE/DEAD Viability/Cytotoxicity kit (Invitrogen) as described above”. Described above where? I cannot seem to find such description. Same goes for similar comment in lines 225/226. Regarding the flow cytometry experiment, it was not clear how the uptake of TiO2 NPs impact size and granularity to an extent that that can be used as a marker for cellular uptake. Also regarding flow cytometry, in line 141/142: “Cellular uptake of 10 µg/ml 100 nm TiO2 PEG NPs was very low, indicating that most of TiO2 PEG NPs were outside the cells or on the cells surface”. How can the authors distinguish there are NPs outside the cells or on their surface? In the Localization of P-gp by immunofluorescence mechanism, this reviewer suggests including the brightfield image of the cells, even if not as an overlay, for reference of the state of the cell during imaging. Additionally, how can the authors distinguish the P-gp on the membrane and internalized by the cell without doing any Z-stack? Besides, that is a qualitative technique and quantitative ones should be performed for more accurate conclusion.

Author Response

Dear Editor,

Thank you for your comments. Please find the revised manuscript on the web site. Revisions have been made after carefully considering the comments raised by the reviewers.

Reviewer 1

The authors should describe the methods more completely. It is often hard to get the full information that would allow to reproduce the experiments. One example is item 4.2, where they state preparation of TiO2 PEG NP was described previously. They should describe the preparation briefly. Item 4.4 presents the same problem mentioned in comment 1. Line 222 says “assessed using a LIVE/DEAD Viability/Cytotoxicity kit (Invitrogen) as described above”. Described above where? I cannot seem to find such description. Same goes for similar comment in lines 225/226.

Answer: According to reviewer’s indication, we have added more detailed information in “Materials and Methods”. (For item 4.2, lanes 217 to 222. And for item 4.4, lanes 233 to 249.) We have also changed the title of item 4.4 at lane 232 to be clearer.

Regarding the flow cytometry experiment, it was not clear how the uptake of TiO2NPs impact size and granularity to an extent that that can be used as a marker for cellular uptake.

Answer: We have clarified evaluation of cellular uptake of NPs in more detail in lanes 265 to 269.

Also regarding flow cytometry, in line 141/142: “Cellular uptake of 10 µg/ml 100 nm TiO2PEG NPs was very low, indicating that most of TiO2PEG NPs were outside the cells or on the cells surface”. How can the authors distinguish there are NPs outside the cells or on their surface?

Answer: According to cellular uptake results, few cells have taken up the NPs. This means most of nanoparticles were outside the cells. The free nanoparticles on the culture medium could not affect the cells response, however, the NPs that interacted with the cell surface could interact with cell membrane receptors and induce change in cellular response. We have added this explanation in the text in lanes 141 to 144.  

In the Localization of P-gp by immunofluorescence mechanism, this reviewer suggests including the brightfield image of the cells, even if not as an overlay, for reference of the state of the cell during imaging. Additionally, how can the authors distinguish the P-gp on the membrane and internalized by the cell without doing any Z-stack? Besides, that is a qualitative technique and quantitative ones should be performed for more accurate conclusion. 

Answer: We took several pictures with different focus. These photos showed the same results, indicating these were shown distinguish the P-gp on the membrane and internalized by the cell. We have added this information on text in lane 160. We have also added calculated fluorescence intensity of p-gp in figure 4C and D. According to this changing, we have added results and figure legends for figure 4C and D in text.

We trust that these changes have addressed all of the reviewers’ comments. We hope that our revised manuscript is now suitable for publication.

Sincerely,

Akiyoshi TaniguchiPhD

Cellular fanctional Nanobiomaterials Group, Research Center for Functional Materials,

National Institute for Materials Science

1-1, Namiki, Tsukuba, Ibaraki 305-0044 Japan

Reviewer 2 Report

The manuscript described a detailed well designed study of the effect of TiO2 PEG NPs on cisplatin cytotoxicity on two types of cancer cells: HepG2 cells, which express P-gp and A431, which show low expression of P-gp. The work is well described and extensive and the results add knowledge to the field, showing that low concentrations of TiO2 PEG NPs increase cisplatin toxicity only towards the cells expressing P-gp by inhibiting P-gp function in these cancer cells. By this mechanism, the MDR is bypassed. The paper is well written and can be published in the IJMS.

Author Response

Dear Editor,

Thank you for your comments. Please find the revised manuscript on the web site. Revisions have been made after carefully considering the comments raised by the reviewers.

Reviewer 2

The manuscript described a detailed well designed study of the effect of TiO2 PEG NPs on cisplatin cytotoxicity on two types of cancer cells: HepG2 cells, which express P-gp and A431, which show low expression of P-gp. The work is well described and extensive and the results add knowledge to the field, showing that low concentrations of TiO2 PEG NPs increase cisplatin toxicity only towards the cells expressing P-gp by inhibiting P-gp function in these cancer cells. By this mechanism, the MDR is bypassed. The paper is well written and can be published in the IJMS.

Answer: Thank you for your comments.

We trust that these changes have addressed all of the reviewers’ comments. We hope that our revised manuscript is now suitable for publication.

Sincerely,

Akiyoshi TaniguchiPhD

Cellular fanctional Nanobiomaterials Group, Research Center for Functional Materials,

National Institute for Materials Science

1-1, Namiki, Tsukuba, Ibaraki 305-0044 Japan

Round 2

Reviewer 1 Report

Dear authors,

thank you for your responses and clarifications to the comments for the paper entitled "The effects of TiO2 nanoparticles on cisplatin cytotoxicity in cancer cell lines". Most concerns were addressed in the revised manuscript. Given the additional information included in the manuscript, the results are more clear, which improved the quality and understanding of the paper. The paper, in the current form, is suitable for publication in the IJMS.